# An Insight into Indonesia’s Challenges in Implementing Newborn Screening Programs and Their Future Implications

**DOI:** 10.3390/children10071216

**Published:** 2023-07-13

**Authors:** Gilbert Sterling Octavius, Vamela Adman Daleni, Yulita Delfia Sari Sagala

**Affiliations:** 1Department of Pediatrics, Universitas Pelita Harapan, Tangerang 15811, Indonesia; 2St. Theresia Hospital, Jambi 36123, Indonesia

**Keywords:** Indonesia, newborn screening, challenges, future implications, congenital hypothyroidism, congenital adrenal hyperplasia, critical congenital heart defects

## Abstract

Due to high entry barriers, countries might find it daunting to implement the NBS program, especially those just trying to start it. This review aims to discuss Indonesia’s barriers that hinder newborn screening (NBS) implementation while discussing the future implications. Literature in Pubmed and Google Scholar was scoured with keywords such as “Newborn Screening”, “Neonatal Screening”, “Indonesia”, “Asia Pacific”, “Barriers”, and “Challenges”. We also searched for relevant references in those published articles. Grey literature, such as state regulations, informative webinars on the topics by experts regarding current situations, and press releases by the Indonesian Minister of Health (MoH), was also searched. Newborn screening is no longer considered just a laboratory test but an array of well-harmonized systems that must be orchestrated well. Some of the barriers Indonesia faces in implementing NBS are a lack of prevalence data, ethical issues, infrastructure, cost-benefit analysis, logistical issues, government support, patient issues, a lack of commitments, and a lack of healthcare workers, specialization, and training. Government support with professional advocates and support groups, proper infrastructure, and a single-payer system for NBS programs are necessary to accelerate NBS programs in Indonesia.

## 1. Introduction

Newborn screening is progressing rapidly, especially regarding laboratory equipment and diagnostic methods. Due to high entry barriers, countries might find it daunting to start implementing the NBS program. Hence, it comes as no surprise that many low- and middle-income countries still struggle to provide some, if not all, universal NBS. Bhutan, Brunei, Burma, Cambodia, East Timor, Indonesia, Laos, Albania, Lebanon, Nepal, North Korea, Tajikistan, Pakistan, Papua New Guinea, the Polynesian Islands, and Kosovo are some countries belonging to the list [1,2,3,4]. A recent article even mentions that NBS alone may not be adequate to prevent adverse events in pediatric congenital hypothyroidism (CH). Many factors are at play when a country tries to implement a successful and sustainable NBS program [5].

Some factors contributing to the challenges may be similar in all other countries, but some aspects will be unique to a specific country. Hence, this review aims to discuss Indonesia’s barriers that hinder the NBS implementation while discussing the future implications. Not only will this review focus on the current challenges in the existing NBS program, which is CH screening, but this review will also focus on why screening for certain disorders such as critical congenital heart disease (CCHD), hearing loss, and congenital adrenal hyperplasia (CAH) is still yet to be implemented universally in Indonesia.

## 2. Materials and Methods

Literature in Pubmed and Google Scholar was scoured with keywords such as “Newborn Screening, “Neonatal Screening”, “Indonesia”, “Asia Pacific”, “Barriers”, and “Challenges”. In order to ensure literature saturation, relevant references in published articles were also searched. Grey literature, such as state regulations, informative webinars on the topics by experts regarding current situations, and press releases by the Indonesian Minister of Health (MoH), was also searched.

## 3. Results and Discussion

### 3.1. Brief Description of Healthcare Profiles in Indonesia

Indonesia is the fourth-largest nation with 273.8 million people, a 0.64% population growth rate compared to 2021. It is also the biggest archipelago in the world, with 17,500 islands spread across the equator. The 2020 crude birth rate was 17.07 births per 1000 people, a 1.55% decline from 2019. Indonesia has a substantial maternal mortality rate of 189 deaths per 100,000 live births. This number is roughly 2.5 times higher than the targeted 70 maternal deaths per 100,000 live births by 2030. Meanwhile, the fertility rate and infant mortality rate stood at 2.28 births per woman and 17.7 deaths per 1000 live births in 2020, respectively [1,2].

In contrast to most other nations, where skilled birth care is only available in hospitals or health centers, Indonesia is one of the few nations that has worked to enhance its maternal health system by attempting to provide competent delivery care in communities and homes. Thousands of village-based midwives who provide reproductive, maternity, and child health services, frequently including giving birth at home or in village clinics, are the foundation of the Indonesian system [3]. These Indonesian village midwives offer public primary health care throughout the nation, along with a variety of neighborhood-level basic healthcare facilities known as “puskesmas” (sub-district health center—daily clinic or inpatient with basic surgery), “pustu” (village health center—usually rural with daily clinic), and “posyandu” (typically once a month health services delivered in small/remote villages). Public hospitals in every district and province support these centers as referral institutions, and local governments are given administrative control. Numerous Indonesian midwives also operate in their own or other private practices, adding to the country’s sizable private healthcare industry. Approximately one-third of hospitals are controlled by for-profit corporations, and many medical professionals split their time between private practices and public institutions. Due to this growing privatization, the rich and poor, as well as urban and rural populations, have huge gaps in access to healthcare services and health outcomes [4]. As a result, only 55.2% of mothers seek assistance with birthing from healthcare providers such as general practitioners and practicing midwives, hospitals (whether state- or privately-run), maternity homes, health centers, or auxiliary health centers [5]. This number means that only 6005 babies were born with the assistance of healthcare providers out of 10,880 births in 2020.

Currently, only CH is a mandatory disease to be screened for in Indonesian newborns. In 2014, the Indonesian Ministry of Health passed a decree mandating all states to conduct routine CH screening for all newborns. However, the cost was shifted towards states and individuals as this program was not yet funded by national health insurance or integrated into national health programs at that time [6,7].

### 3.2. Challenges of Implementing NBS

#### 3.2.1. Lack of Prevalence Data

Epidemiological data guides the government on where and how to tackle a disease entity best. This data is necessary because epidemiology varies greatly, even within one nation. Other studies have shown that geographical variation in a country exists. For example, the prevalence of CH was higher in the coastal area of China (4.73 newborns per 10,000) as compared to the inland area (4.16 newborns per 10,000) and remote area (2.58 newborns per 10,000) [8]. In Indonesia, the prevalence of CH is also likely influenced by the geographic area, as the incidence is higher in iodine-deficient countries than in non-iodine-deficient countries [9]. However, studies are sparse and concentrated in selected cities only. Furthermore, only CH and CAH pilot studies had been conducted for a prevalence study.

Epidemiologic studies that analyze the contribution of preterm births, iodine deficiency or excess, consanguinity, genetic factors, and living location are seriously needed to analyze the distributions and identify potential risk factors of CH and other identifiable diseases screened by NBS programs across Indonesian regions. Other countries that are just starting their NBS programs are also struggling with data gathering, emphasizing that prevalence data is the pivotal starting point for an NBS [10,11].

However, even though pilot studies have been conducted, uptake can sometimes still be slow or nonexistent. It took 13 years before the first nationwide mandate was officially issued to screen for CH in all Indonesian newborns. Meanwhile, CAH NBS is still not mandatory, even though a pilot study has been performed [6]. The lack of prevalence data may not be the biggest problem to jump-starting an NBS program. Other reasons, such as those delineated below, may contribute to the challenges of implementing the NBS program.

#### 3.2.2. Economic Issues

The costs of NBS can be influenced by costs related to the acquisition of machines or equipment needed for screening, consumables related to screening, the salary of the technicians or primary health care (PHC) workers, administration tasks, and hardware database [12]. However, these same reasons are also why the cost-benefit ratio is not straightforward to calculate, as estimating the downstream costs requires long-term follow-up [13]. For example, it is estimated that if a CAH assay costs $26 per sample and the result is a false-positive, the high financial and emotional burden costs are estimated at more than $800 per infant [14]. Another example of why cost-benefit analysis may not seem straightforward is the higher cost of two-tier CAH screening compared to one-tier screening. Despite the higher costs, recall rates are lower, and positive predictive values are better for two-tier CAH screening, which makes it more desirable [15]. Lastly, the timing and case management may also impact the cost-benefit of screening as mortality may remain high, thus blunting the positive impact of screening [16].

The lack of cost-benefit analysis in Indonesia further compounds the support for NBS in Indonesia. Without analysis, the health-technology assessment (HTA) could not be done. Hence, no recommendations will be in place to support the NBS program in Indonesia. For example, one study in Italy that conducted a universal newborn hearing screening program found that the financial investment could be recouped in two years [12]. However, should Indonesia move forward with newborn screening without comprehensive HTA, one study found that even with pessimistic assumptions, NBS seemed to be highly cost-effective. Over the long term, funding a comprehensive NBS program would likely save money for society [17]. Introducing a comprehensive tracking and follow-up program, rather than just enhancing laboratory equipment, is the best way to implement a new screening technique [18].

#### 3.2.3. Ethical Issues

Ethical principles for NBS programs should follow the following guidance: (1) Respect for parents while assuring autonomy and allowing parents to choose for themselves after acquiring thorough and complete knowledge; (2) Protecting privacy in organizing and delivering information and care; (3) Beneficence with a favorable benefit-risk relationship; (4) Non-malfeasance through efficient, effective program implementation; and (5) Justice for all people involved with equitable deliverable care [19].

As per national regulation number 78 in 2014 in Indonesia, specific informed consent forms for CH screening are not required, and the consents can be grouped with other consents for routine newborn care. Parents who refuse the screening or confirmatory testing must sign a specific refusal consent form to prevent malpractice lawsuits [20]. However, consent is still an issue, as some parents disagree with repeated blood samples being taken [6]. In some places, the informed consent form also did not exist or was explained in passing [21].

More countries are now moving away from obtaining consent and instead choosing an “opt-out” policy [22]. However, some ethical issues are brought up due to the number of false positives introduced during screening. The false-positive results could cause anxiety in the family, stress, a sense of guilt, impatience, a frantic search for unnecessary aid, and depression [12]. There is also a suggestion for implementing a policy on giving full disclosure, withholding information, or letting parents decide on which information to give them. These three choices have their benefits, drawbacks, and controversies [19]. Lastly, false-positive results may wrongly influence parents to think that their children are ill and, therefore, more prone to other diseases, a condition termed vulnerable child syndrome [23].

The debate on the ethics of newborn screening even extends to religious freedom, parental discretion, and the child’s rights. On one side of the argument, parents can refuse screening if their decision is based on sincerely held religious beliefs [24]. However, another view is that the absence of screening may lead to death, which, considering the little harm caused by NBS, far outweighs the benefits; NBS still should be imposed on all infants even if the parents have their own personal or religious beliefs [25].

Lastly, there is an ethical concern about sample storage. In Indonesia, there are still no clear regulations on sample storage, and there is a growing debate on whether informed consent is required for sample storage, as some may argue that clinicians are violating human rights without informed consent. The other side of the argument is that, from a public health perspective, storage may promote better health and quality of life, and hence no informed consent is required [26].

#### 3.2.4. Lack of Infrastructure

Newborn screening cannot save as many lives as possible if precise diagnoses, secure transportation, and high-quality medical, surgical, and interventional therapies are not readily available. The lack of infrastructure cannot be alleviated easily, as other factors such as competing priorities, poor structural organizations, a lack of financial resources, a lack of trained human resources, and educational infrastructure also affect the decision to build a particular infrastructure [27]. Hence, instead of considering investing in new infrastructure, it is essential for policy implementation to include an assessment of the existing infrastructure to ensure the effective utilization of all resources [28].

There are eleven CH screening laboratories situated on every major island in Indonesia. Countries with a larger population generally have more laboratories, such as Russia, which has 78 laboratories, while many European countries may only have one or two. One study found that one European laboratory may handle 100–20,000 newborn screenings. Hence, with a total of 4,496,383 newborns being born in Indonesia in 2021, almost 225 laboratories are needed to accommodate NBS in Indonesia. While this number seems unattainable, the government needs to tackle the lack of laboratories with an efficient design of the NBS network [29].

The availability of screening tools necessitates countries to improvise or use whatever resources they have for screening if anything is available. For example, when developed countries used a combination of screening procedures for infant hearing loss, some hospitals in Indonesia still used otoacoustic emission (OAE) only [30]. It should be noted that there are no mandatory rules for infant hearing loss screening in Indonesia.

In Indonesia, there are only 40 hospitals that can provide a catheterization laboratory service, and only ten hospitals can perform open heart surgeries. Indonesia still needs 1282 cardiovascular and other specialists to care for its people. The Ministry of Health estimates that 12,000 babies suffer from congestive heart failure (CHF) due to long-standing and uncorrected CHD, and only half are being treated [31]. Hence, it is common to see adults with uncorrected congenital cardiac malformations [32].

Included in the infrastructure is the information technology (IT) system. The lack of proper IT systems in Indonesia greatly hampered data collection. Indonesia’s data collection system is still largely individualized based on the hospital registry. When a nationwide attempt is made to homogenize the data, many new problems appear, such as inconsistencies in coding the diagnosis, a lack of infrastructure, and a lack of follow-up and feedback for the projects [33]. Numerous screening programs suffer from a lack of regional databases and consistent data collection. The lost-to-follow-up rate is frequently high or unknown without tracking in less-developed countries with less-developed NBS programs [30]. In Indonesia, the data collection relies on individual PHC data as the person in charge inputs the data. Without proper verification, systematic mistakes and bias may occur during data collection, which greatly impacts the available data. It is also important to note that the NBS’s online data collection is currently combined with other programs, such as the infant mortality rate, making data collection difficult.

#### 3.2.5. Logistical Issues

The Indonesian Ministry of Health released a policy in 2022 stating that all PHCs and hospitals would send the screening cards to the designated referral laboratories. The needed logistics (screening cards, drying racks, plastics, and educational materials) to conduct the screening are also obtained from these referral laboratories [34]. All PHCs and hospitals need to sign a cooperation agreement that lists the rights and obligations of both parties in conducting the NBS program. Since these 11 laboratories handle the logistical supply chain for all PHCs and hospitals, the distribution and procurement of the necessary equipment are often late [35]. The massive archipelago also compounds delivery logistics in Indonesia, as shipments may be delayed due to numerous factors such as weather conditions, public holidays, and delivery courier capacity. This logistical issue results in a variable supply of screening cards, affecting the NBS program’s consistency [21]. Some facilities also encourage batching or accumulating specimens to increase efficiency in sending the samples. However, this practice has been shown to waste precious time and may delay the diagnosis and treatment of a child, potentially endangering their lives [36]. Furthermore, besides being an archipelago, the uneven population distribution is also a major factor to be considered, as some accessible countries are very densely populated (such as Jakarta). In contrast, faraway countries, such as those outside Papua’s capital city, have a very low population density. This population density issue further compounds Indonesia’s logistical issues and supply chain.

As a result, the turn-over time from taking the sample to knowing the result may take as long as 7–14 days. Communication is also an issue if the central laboratory does not reach back; it is assumed that no positive cases are detected. Hospitals and PHCs may get the screening cards back after one year [21]. Furthermore, there are still no policies or regulations on screening card storage or disposal.

#### 3.2.6. Government Support

The need for government support is prevalent in the literature [10,37,38]. India, one of the Asian countries that also struggled to implement NBS, lacks government support. As a result, their NBS programs were conducted through individual efforts [39].

The lack of support translates partially into varying results in practice. One study in 2016 found that those who paid out-of-pocket (OOP) for CH screening had better sampling timeliness than those whom the government covered. However, samples that were government-funded had better timeliness in sending them as compared to OOP [40].

The Indian Council of Medical Research (ICMR) suggested that good coverage might be ensured if the HCWs were given a small incentive and trained to collect blood samples [22]. Based on previous experiences, giving an incentive may boost short-term screening performances but will backfire once the subsidies for incentives are stopped. Hence, we argue against giving an incentive, but the government needs to improve the healthcare system for the smooth running of the NBS program.

In Indonesia, CH screening is the only commitment to an NBS program. The funds needed to operate the CH NBS program came from regional income and expenditures and operational health budgets. There was discord where the government felt the fund was adequate for screening, but the PHC could only achieve 50% of the targeted CH NBS [21]. There was a revision in the Ministry of Health mandate to accelerate CH screening in newborns in 2020 and 2021. In 2022, the Ministry of Health accelerated the NBS program for CH with 11 referral laboratories. They targeted 463,000 screened samples, or 10% of all newborns. Until the end of 2022, only 99,263 samples (21.4%) were screened from the initial target. However, besides CH screening, the absence of government support for NBS programs has been palpable.

#### 3.2.7. Lack of Healthcare Workers, Specialization, and Training

Good-quality dried blood samples are needed to ensure the results are trustworthy. However, when CH screening was re-launched by the Indonesian government, there was a lack of re-training for all healthcare workers (HCWs), including midwives and nurses. Without continuous training or at least a refresher course, the new batch of freshly graduated HCWs will not feel confident about collecting proper blood samples [6,34,41]. Unsurprisingly, many samples were discarded due to improper collecting techniques. The first pilot study in Indonesia found that 0.78% (out of 6797) of filter paper spot samples were not analyzed because the samples did not meet the required criteria. The authors suggested that constant supervision may be needed during sample collection, which may not be feasible in the field [9]. Constant supervision is needed, as not all HCWs are trained to collect blood spot samples. Some HCWs learned the technique from others who attended the training and not directly from the experts. Workshops to train HCWs to collect bloodspot samples are not regularly conducted, and only certain big cities conduct such workshops. Hence, not all HCWs will be adequately trained. Furthermore, multiple attempts may be needed to collect proper samples if the HCWs are not properly trained. Parents may get the wrong impression that multiple punctures are needed, which might cause future refusals.

Hospitals and other PHCs also lack standard operating procedures (SOP) to conduct the CH NBS [21]. Without a proper protocol, HCWs will individually have their own methods and preferences for conducting the NBS. Hence, much variability will be introduced, impacting the resources needed to conduct this NBS. A proper SOP will reduce unnecessary variability, streamline the process, and even reduce the need for extra human resources. Staffing shortages are also a real issue that hinders the implementation of NBS. Without proper workload sharing, HCWs will likely not have time for newborn screening, thus leading to more errors [28,42].

Primary care physicians are expected to share test results with families, offer guidance on the significance of a positive result and the suspected condition, make sure confirmatory testing is carried out, assist families in navigating the healthcare system by making the proper referrals, and track the condition’s health outcomes throughout childhood. However, one study found that many primary care physicians lack the necessary skills to handle the follow-up care of children with positive newborn screenings, including the first guidance, diagnosis, and referral to a specialist [43].

In Indonesia, other than infrastructure issues, correctable surgeries for CCHD are not easily performed due to the absence of trained congenital heart surgeons, a lack of facilities, limited human resources, and severe conditions at the onset of diagnosis [44]. In 2007, Indonesia only had 25 pediatric cardiologists, nine in the capital city, and three pediatric cardiac surgeons [32]. In addition to the lack of HCWs, the distribution across the archipelago is also uneven, creating a relative lack of HCWs in rural areas.

Indonesia lacks geneticists or counselors since genetic diseases have not been considered serious in Indonesia [45,46]. Genetic services are also sparse and only available in selected tertiary hospitals [46]. The absence is significant because some NBS disorders are heavily linked to genetic mutations, such as CCHD [47] and hearing loss [48]. Counselors are needed for genetic counseling, as a study has shown that genetic counseling after identification of carrier status reduces the mean anxiety levels [49]. This is especially true when Indonesia is a country with unknown data but presumably moderate levels of consanguineous marriage [50].

#### 3.2.8. Lack of Commitments

Since CH screening became a mandatory national program, all PHCs and hospitals need to adopt and support this program. Furthermore, the Indonesian hospital accreditation board requires all hospitals to achieve 100% completion in executing national programs. However, based on our experience, some centers will do the minimum required by the government to pass the criteria. Hospitals lack an overall commitment to contact and track newborns with positive screen results, as it only increases their costs and human resources burden [6]. Furthermore, due to the lack of HCWs or commitments, data collection and submission to the government is often late or incomplete.

#### 3.2.9. Patient Issues

As a massive archipelago, getting from one country to another may take time and money. There is also an indirect cost in the form of a loss of daily income. Since most Indonesians belong to the middle class and lower, traveling costs and loss of daily income have become significant [1]. Hence, it is not surprising that only three came when one pilot study recalled ten patients for CH and CAH confirmatory tests [6]. However, this complicates the signing of refusal consent forms. If the parents did not show up for confirmatory testing, one could not assume that they refused the tests since multiple factors contributed to their absence. Parents initially consented to the screening tests but not to the confirmatory tests, as they assumed their children would be normal. One study in Indonesia found that 22.8% of patients (out of 378 patients) with suspected CCHD referred from primary and secondary care centers were lost to follow-up [51].

Even if hospitals and other PHCs commit to recalling the patients, tracking the address or calling back with the provided phone numbers proves challenging since both pieces of information are incorrect or no longer valid [9,52]. The information taken from the national identity card could also be misleading, as the patients could come from other cities, have moved out, or have changed their phone numbers [52].

Parents also lack the necessary information about the importance of NBS. Ideally, parents should be educated by HCWs in the late trimesters about NBS and why it is important. After the consultation, parents will have the time to research and ponder the pros and cons. However, antenatal consultation is still not done, as the obstetricians believe it is the pediatrician’s job, while pediatricians only meet the parents after the baby is born. The lack of information media to inform parents of the importance of NBS also compounds the problem, as parents are unsure where to get reliable information [21].

### 3.3. Future Implications

#### 3.3.1. Support from the Government and Professional Advocates

The government is responsible for leadership, establishing national policies to lower the barriers to NBS success, working with third parties and other stakeholders, raising public awareness, enacting legislation to formalize the NBS operation, and collecting data and tracking the systems [53]. By setting national goals, the government could help lay out a blueprint with its target to meet the NBS goal [54].

Including more support groups and formal organizations in the advocacy professional system will strengthen and expand the NBS program. Another HCW that could be included in this group is a midwife. They seem to be the best healthcare professionals to educate about NBS in the third trimester in Indonesia. This finding is also in concordance with a survey in Europe, where midwives seem to be the most appropriate choice [55].

Professional advocacy groups may also play a role in helping to harmonize the screening procedures. For example, although dried blood spot collections are preferable after 72 h, professional advocacy groups may release a national guideline on circumventing this problem. Another example will be the harmonization of cut-off levels, as different countries in the same nation may have different machines, policies, and standards [56].

There is a need to lower the barrier to follow-up. For the NBS program to succeed, detection is not the sole factor. Diagnostic confirmation, treatment, and long-term follow-up are as necessary as the screening. However, factors such as distance from the hospital, transportation challenges, worry and uncertainty about the referral hospital, procedural issues, ignorance and lack of understanding, and inadequate visibility and service availability are all factors that might contribute to patient distress, which ultimately leads to parents not showing up for follow-up appointments [30].

Before implementation, it is essential to thoroughly investigate Indonesia’s birth delivery procedure and who would be best qualified to conduct the test. This process entails having a thorough awareness of the typical practice areas and training requirements of the healthcare professionals who will be giving the test [28]. The lack of professionals may also hamper the success of NBS in Indonesia, as follow-up requires a multidisciplinary team (the specialists, experienced nursing staff, and technicians) [57].

The budget allocation must also include the treatment options required for the infants. For example, hearing loss will require an external hearing aid, which may be expensive for the unfortunate [48]. Hence, the government must allocate its spending on screening, treatment, and follow-up.

Establishing a national committee for newborn screening will centralize and coordinate NBS better. By coordinating NBS centrally, performing quality assurance, and monitoring the effectiveness of the NBS program, this committee will improve NBS in Indonesia by ensuring a high-quality, high-volume, and highly coordinated program with a low return-to-follow-up [53]. One major problem is the discordance between national health insurance policies and the current screening standard. Collecting heel-prick samples around 48–72 h after delivery is advisable to screen for CH. However, Indonesia’s national health insurance stated that all healthy babies should be discharged before 48 h, sometimes even before 24 h, making screening difficult.

This dilemma is not unique to Indonesia. Other countries also experience early newborn discharge, necessitating blood specimens to be collected before 24 h of life [58,59]. If the newborn needs to be discharged before the recommended specimen collection, it may be better to collect the samples and perform a second screening test if indicated [60,61,62]. Some guidelines allow sample collection on day one if the infant is discharged from the facilities, with repeated sampling on days 3–5 [36]. Early sampling is therefore justified in healthcare systems where ensuring newborns return after 48 h may be challenging [63]. While this is ideal, the situation in Indonesia may not allow this, as parents may be reluctant to return for a second test due to distance, financial issues, and time. However, this hypothesis needs to be confirmed and addressed in future studies. Hence, the government needs to work with all stakeholders and professional advocates to look at all the available options and tackle this problem.

The government may also look at other countries’ best practice models in conducting NBS, such as the law mandate for NBS in Italy and France, NBS infrastructure reorganization and same-day, second-tier testing results in Italy, a nationwide IT system for NBS in Slovenia, a clear hierarchical structure of advisory organs to the Ministry of Health in France, pilot studies as a pivotal starting point to start a new NBS program in Germany and the Czech Republic, as well as a national biobank for dried blood spots storage in Sweden [38].

Lastly, the government needs to ensure the quality of NBS programs. The indicators can be the percentage of eligible infants screened for a certain disorder, the number of infants with positive screening, the number of infants with certain conditions identified by newborn screening, the number of newborns with a specific disorder who were missed on the NBS, and the timing of screening activities [64]. One systematic review laid out a checklist that can help the government and stakeholders track the quality of the NBS process and make necessary adjustments to the program based on the quality [65].

#### 3.3.2. Single-Payer System

Insurance coverage for children with the affected disorders is crucial for exploiting the health benefits of the screening since relying on out-of-pocket payments will be difficult and public insurance will not cover these disorders [16]. The coverage must include drugs, surgeries, and aftercare [66]. The care needed for some cases, such as CCHD, is too expensive for the average Indonesian family to cover, not including the travel expenses to get the service and the income loss from work leave [32].

However, relying on national insurance may take some time as it struggles with financial sustainability, the “missing middle”, an equity gap in insurance coverage, and challenges in service preparedness. Indonesia needs to invest in proper maternal care to enhance fetal growth and development, which would reduce the burden of neonatal and infant mortality and long-term risk for non-communicable diseases, thereby reducing the national insurance burden. Ironically, this means spending more budgets on NBS programs. Thus, the Indonesian government is likely faced with a catch-22 situation [67].

#### 3.3.3. Upgrading the Information Technology

The role of information technology in the NBS program is increasing. Seven critical modules in an information management system will enhance NBS programs, such as patient record management, laboratory information system with quality control, clinical or medical review, case management, sample lifecycle management, reporting and analytics, and decision support [68]. Data collection, transfer, and analysis contribute to increased knowledge of cut-off data and the prevalence of a specific disorder in a certain region. Ultimately, clinicians who utilize this data will provide better care for the patient [69]. The role of electronic health records (EHR) is undeniable in providing reliable and accurate information and administering quality health care [70]. Indonesia is mandating all hospitals and PHCs to implement a centralized EHR by December 2023, but there is still a long way to go.

Rustama reported that missed cases could not be easily identified as there is no reporting system for affected infants missed by the programs [9]. He also mentioned that infrastructure must improve, the recall system must be aggressive, education and information campaigns for parents and medical professionals must be intensified, and the Ministry of Health must be persuaded to give a national mandate [9]. Since then, despite the national mandate, Indonesia’s NBS has been progressing slowly, emphasizing the importance of good infrastructure.

## 4. Conclusions

Newborn screening (NBS) is not easily implementable in other countries because it requires a system. The sustainability of a system depends on the seamless integration of all system parts within the limitations of regional political, economic, and geographic restrictions. Padilla and Therell mentioned six essential elements for an NBS program to succeed: education, screening, early follow-up, diagnosis, management, and evaluation [71]. The success of the whole NBS system depends on collaboration and coordination with academic institutions and private partners (confirmatory laboratories, medical facilities, third-party payers, and other non-governmental organizations).

Others argue that to guarantee that NBS programs operate effectively and to the highest standards of quality, they require a clearly defined structural organization or policy framework with appropriate guidelines or recommendations [55]. NBS is a system rather than just a simple laboratory test. This system necessitates the seamless integration of genetic counseling, treatment, and follow-up, as well as medical and public education, sample collection, transportation, screening, and confirmation. As a result, it is becoming increasingly clear that NBS would be best handled as a program for general public health. Its success will depend on a wide range of parties exchanging knowledge and working together, such as the government, healthcare providers, professional associations, patient advocates, and other civil society organizations (Figure 1) [72].

There is no particular standout problem that causes the slow progress of NBS. Instead, all the problems mentioned above contributed to the reason why NBS programs are implemented slowly. Newborn screening is a public health system that requires the connectivity of many different components. The lack or absence of one element will derail the success of an NBS program. Indonesia must prioritize and tackle the barriers to implementing NBS programs to conduct this program successfully. The government has taken a step forward by mandating compulsory CH screening. The advancement of NBS programs in Indonesia will depend on whether all the problems mentioned above are solved simultaneously. However, since this is a review, we cannot point out which factor is the most dominant, if any. Hence, further research and involvement with all stakeholders are necessary to tackle this issue.

## Figures and Tables

**Figure 1 children-10-01216-f001:**
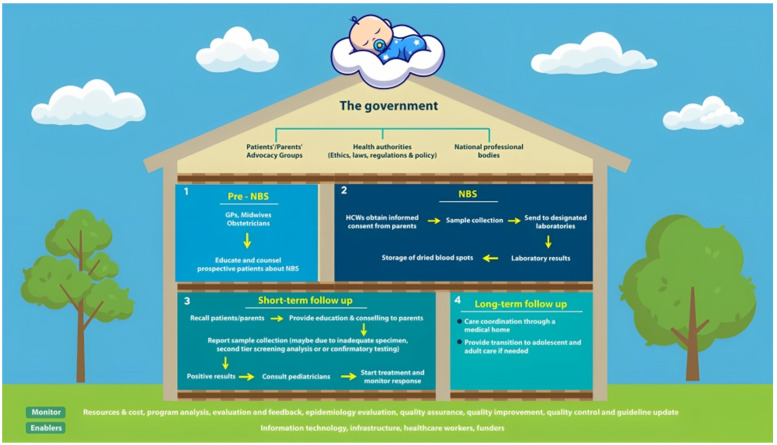
The newborn screening system is a system of interconnected public health policies.

## Data Availability

Not applicable.

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
