# Peer review of "An Insight into Indonesia’s Challenges in Implementing Newborn Screening Programs and Their Future Implications"

_children, 2023, doi:10.3390/children10071216_

Round 1
Reviewer 1 Report (New Reviewer)
It is useful to describe the barriers to newborn screening in a large, low to middle income country but this report is difficult to follow.
1. The authors should consider describing the current situation in Indonesia early on to set the scene for readers eg/ number of annual births, maternity system and proportion of hospital births (if known), any existing newborn screening in place, ie bloodspot, deafness or pulse oximetry.
2. The manuscript seems to be focused on bloodspot screening but this is not stated. The authors sometimes mention hearing and pulse oximetry screening too - the scope needs to be clarified early in the paper and then followed in a consistent way.
3. The introduction is particularly difficult to follow and should be revised so that ideas within a paragraph are linked and only the most relevant information is given. For example, it is true that NBS is a system than just a laboratory test, and that is linked to why NBS alone cannot guarantee improved outcomes for CH, as these also depend on medicine availability and high quality follow-up care. Why are academic partners and private institution considered essential to the success of a NBS programme?
4. Why is a lack of prevalence data a barrier to establishing newborn screening? Why isn't CH and CAH prevalence data from the small Indonesian pilot cited sufficient, in combination with prevalence data from similar populations. Many programmes would start this way and plan to gather more detailed epidemiological data over time.
5. Lack of infrastructure. This section would be easier to follow if it were re-organised. Why are hearing testing and cardiac catheterisation facilities discussed before the infrastructure required for bloodspot screening? Do the authors mean congestive heart failure or something else?
6. Logistic issues. The authors should consider emphasising the very large population here.
7. Government support. It is not clear what government support is available or has been pledged to support newborn screening.
8. Lack of HCW and training. I don't understand the recommendation for constant supervision - in other jurisdictions bloodspot collection is taught as a skill that HCW then become proficient in. Scope of this manuscript - if it is about bloodspot screening only, the lack of cardiologists is not relevant.
9. Line 273. suggests that access to CH screening is mandatory in Indonesia - if this is the case this information needs to be provided much earlier in the manuscript to help set the scene for readers. And does this mean that bloodspot screening would initially be for CH only? There is logic to starting with CH as compared with more time-critical disorders whilst the infrastructure is established, this could be expanded upon in the "going forward" section.
10. Lines 314-315 - what do the authors mean about a trend of including obstetricians in the diagnosis of Down syndrome?
11. Sample collection time is often 48-72 hours, however the optimal time depends on the screening goals. Some NBS programs eg in Uruguay have started CH screening using cord blood samples. If the initial focus in Indonesia is on detecting severe CH (and CAH) then early bloodspot collection prior to discharge would likely be sufficient. The US review cited suggesting babies who are screened <24 hours return on days 3-5 for a further collection may not be relevant to the Indonesian situation.
This is currently a major limitation - there are a number of occasions where it is hard to know what the authors mean and they are unlikely to mean what is written. Even the title is not grammatically correct. The manuscript needs extensive language revision.
Author Response
Thank you so much for your valuable input. We have revised our manuscript according to the suggestions made by the 1st reviewer, with the details included in our word-by-word response

Reviewer 2 Report (New Reviewer)
The main problem of the work is to discuss the issue in a selected geographical area. The publication itself was prepared correctly, it discusses the basic issues. I believe that it is necessary to
1. expand the limitations of the work
2. make more reference to the world literature.
3. check whether the proportions of the various elements of the work correspond to the requirements of the journal.
Author Response
Thank you so much for your valuable input. We have revised our manuscript according to the suggestions made by the 2nd reviewer, with the details included in our word-by-word response

Round 2
Reviewer 1 Report (New Reviewer)
The manuscript has been much improved.
This manuscript is a resubmission of an earlier submission. The following is a list of the peer review reports and author responses from that submission.
Round 1
Reviewer 1 Report
This manuscript needs to be rewritten in a more academic style.
Authors need to replace phrases such as: "We also searched", "We searched for literature in Pubmed and Google Scholar", etc.
Fig 1 doesn’t look good and the text is unreadable.
Authors need to replace phrases such as: "We also searched", "We searched for literature in Pubmed and Google Scholar", etc.
Author Response
Thank you for your comments. Please see the attached file for our comments.
